# How Is Fitness of *Tribolium castaneum* (Herbst) (Coleoptera: Tenebrionidae) Affected When Different Developmental Stages Are Exposed to Chlorfenapyr?

**DOI:** 10.3390/insects11080542

**Published:** 2020-08-17

**Authors:** Anna Skourti, Nickolas G. Kavallieratos, Nikos E. Papanikolaou

**Affiliations:** 1Laboratory of Agricultural Zoology and Entomology, Department of Crop Science, Agricultural University of Athens, 75 Iera Odos str., 11855 Athens, Attica, Greece; annaskourti@aua.gr (A.S.); nikosp@aua.gr (N.E.P.); 2Directorate of Plant Produce Protection, Greek Ministry of Rural Development and Food, 150 Sygrou Ave., 17671 Athens, Attica, Greece

**Keywords:** chlorfenapyr, red flour beetle, survival, analysis, demography, biological features

## Abstract

**Simple Summary:**

*Tribolium castaneum* is an important pest of stored products. Most studies are focused on the immediate and/or delayed mortality effects, while there are no data on the effects of insecticides on the population fitness. This study deals with the effect of chlorfenapyr on *T. castaneum*, investigating the cost of exposure of different developmental stages on population performance, by using life table statistics and a survival analysis method. For this purpose, eggs, larvae, and parental adult females of *T. castaneum* were exposed to chlorfenapyr and birth or death rates were calculated daily. The exposure of eggs and larvae to chlorfenapyr was detrimental for *T. castaneum* and they did not complete development. When parental females were exposed to chlorfenapyr, the progeny survival curve, biological features, as well as the life table parameters did not differ significantly compared to the control treatment. We expect these results to have bearing on the management of *T. castaneum*, since the repeatedly insecticidal applications could be reduced in storage facilities.

**Abstract:**

*Tribolium castaneum* (Herbst) (Coleoptera: Tenebrionidae) is an important pest of stored products. Insecticidal treatment is a common practice for the control of this notorious insect pest. Most studies are focused on the immediate and/or delayed mortality effects, while there are no data on the effects of insecticides on the population fitness. This study deals with the effect of chlorfenapyr on *T. castaneum*, investigating the cost of exposure of different developmental stages on population performance, by using life table statistics and a survival analysis method. For this purpose, eggs, larvae, and parental adult females of *T. castaneum* were exposed to chlorfenapyr and birth or death rates were calculated daily. The exposure of eggs and larvae to chlorfenapyr was detrimental for *T. castaneum* and they did not complete development. When parental females were exposed to chlorfenapyr, the progeny survival curve, biological features, as well as the life table parameters did not differ significantly compared to the control treatment. Thus, egg hatching, larval and pupal developmental periods, female and male longevities for the control treatment, and the progeny of the females that were exposed to chlorfenapyr were 4.66 and 4.76 days, 25.85 and 25.71 days, 5.00 and 5.26 days, 87.33 and 104.22 days, and 76.87 and 91.87 days, respectively. In addition, the mean values of the net reproductive rate, the intrinsic rate of increase, the mean generation time and the doubling time for the control treatment and the progeny of the parental females which were exposed to chlorfenapyr were 14.3 and 9.3 females/female, 0.038 and 0.028 females/female/day, 1.039 and 1.029, 70.0 and 76.9 days, and 18.5 and 24.9 days, respectively. We expect these results to have bearing on the management of *T. castaneum*, since the repeatedly insecticidal applications could be reduced in storage facilities.

## 1. Introduction

The potential of an insect population to increase through time and space relates closely to some life history traits such as survival, development, and fecundity [1,2,3,4]. In general, high survival rate and fecundity along with short developmental duration are the characteristics that favor the performance of insects [5,6,7]. In order to understand and predict future effects of several abiotic and biotic factors (e.g., temperature, relative humidity, insecticide treatment, and competition) regarding the fitness of insects, the knowledge of how life history traits fluctuate with these factors is of high importance [2,4].

Life table statistics constitute an efficient tool for the evaluation of the potential population increase of insects [4,8,9,10,11,12]. Tabulating survival and reproductive schedules of individuals from birth to death is fundamental for the construction of insect life tables [8]. This results in the calculation of several demographic parameters which are indicative of the potential increase of a population. For instance, the Malthusian parameter, i.e., the intrinsic rate of increase, as well as the finite rate of increase, are basic demographic parameters which allow the assessment of insect population fitness and a basic feature of population models [9,13,14,15]. In addition, the net reproductive rate and the doubling time are also indicators of the future population development of insects [5,7].

The red flour beetle, *Tribolium castaneum* (Herbst) is a serious pest of a large spectrum of stored products. Although this species prefers flours and milled products, it has been also recorded on a variety of cereals and other raw commodities such as legumes, nuts, spices, grains, oilseeds, cottonseeds, spices, dried fruits, pulses, cocoa beans, and processed foods [16,17,18]. *Tribolium castaneum* causes serious quantitative and qualitative losses of stored products throughout the world [19,20,21]. Adults may cause allergic reactions through the release of quinone substances to infested commodities making them unsuitable for consumption [22,23].

The continuous treatments of storage facilities and stored-commodities with insecticides, which aim to reduce insect infestations, lead to the development of resistance phenomena in these organisms [24]. *Tribolium castaneum* is included among those stored-product insects that have globally exhibited resistance to contact insecticides since the previous decades [24,25,26,27,28]. Thus, constant research efforts focus on the development and evaluation of new active ingredients with elevated insecticidal properties [29,30,31,32]. The pyrrole derivative 4-bromo-2-(4-chlorophenyl)-1-ethoxymethyl-5-(trifluoromethyl)pyrrole-3-carbonitrile (chlorfenapyr) is a non-neurotoxic substance that induces oxidative phosphorylation to mitochondria and disrupts the synthesis of adenosine triphosphate (ATP). Chlorfenapyr exhibits low toxicity to mammals and constitutes the only commercialized pyrrole derivative [32,33,34,35,36,37]. So far, it is registered in the USA for crack and crevice treatments where urban pests and stored-product insects may shelter [32,38,39]. Previous studies have well documented that chlorfenapyr is an effective insecticide against adults and/or immature stages of numerous stored-product insects on various types of surfaces, including storage bags, such as *Liposcelis bostrychophila* Badonnel; *Liposcelis entomophila* (Enderlein); *Liposcelis decolor* (Pearman) (Psocoptera: Liposcelididae); the larger grain borer, *Prostephanus truncatus* (Horn) (Coleoptera: Bostrychidae); the lesser grain borer, *Rhyzopertha dominica* (F.) (Coleoptera: Bostrychidae); the rice weevil, *Sitophilus oryzae* (L.) (Coleoptera: Curculionidae); *T. castaneum*; the confused flour beetle, *Tribolium confusum* Jaquelin du Val (Coleoptera: Tenebrionidae); and the khapra beetle, *Trogoderma granarium* Everts (Coleoptera: Dermestidae) [38,39,40,41,42,43,44]. Chlorfenapyr has also been evaluated as grain protectant by causing ≥90.0% mortality to *P. truncatus* adults on maize at 1 ppm while it killed all *S. oryzae* adults at 5 ppm on wheat [45].

Considering the high economic importance of *T. castaneum*, we initiated a study in order to estimate the toxic effect of chlorfenapyr on this pest. Contrary to common approaches, which investigate the immediate and/or delay mortality caused by contact insecticides [38,42,43,44,46], we focus on the lifecycle events related to survival and reproduction. Recently, Stark and Banks [47] used life table analysis to evaluate toxicity data and suggested that time-varying demographic processes are basic tools on the evaluation of the success or failure of insecticidal treatments. In this task, we exposed cohorts of eggs, larvae, and parental females of *T. castaneum* on concrete surfaces treated with chlorfenapyr. Using life table statistics and a survival analysis method, we explored the potential effects of chlorfenapyr on the population fitness. To our knowledge, this is the first study that adopts a demographic and survival analysis approach to investigate the efficacy of chlorfenapyr on insect pests.

## 2. Materials and Methods

### 2.1. Insects

Colonies of *T. castaneum* were multiplied at the Laboratory of Agricultural Zoology and Entomology, Agricultural University of Athens, on white soft wheat flour (variety mixture, made from the endosperm only) at 30 °C, 65% relative humidity and continuous darkness. The founding colony was collected from southern Greece in 2003.

### 2.2. Commodity and Insecticide

Pre-sieved white soft wheat flour (mixture variety made from the endosperm only) that was not infested by pests and not treated with pesticides, was used during the experimental process. The following conciseness of nutrients per 100 g of flour was indicated on the label of the product: 72.6 g carbohydrates, 10.3 g proteins, 1.6 g fiber, 1.1 g fat, and 0.05 g salt. Flour was heated to 50 °C or hydrated with distilled water for adjusting its moisture content to 13.5% as estimated by a calibrated moisture meter (mini GAC plus, Dickey-John Europe S.A.S., Colombes, France) [6,7].

The insecticidal formulation used in the experiments was Phantom EC containing 21.45% chlorfenapyr active ingredient (a.i.) (provided by BASF Hellas, Amaroussion, Greece).

### 2.3. Exposure of Eggs

Tests were conducted in Petri dishes (8 cm diameter by 1.5 cm height) having each 50.27 cm^2^ surface area. The bottoms of dishes were covered by concrete CEM I 52.5 N (Durostick, Aspropyrgos, Greece) 24 h before the initiation of the experimentation. The vertical internal sides of the dishes were covered by polytetrafluoroethylen (60 wt % dispersion in water) (Sigma-Aldrich Chemie GmbH, Taufkirchen, Germany) to block any escape attempts of the insect individuals. Chlorfenapyr was applied at the label dose of 0.11 mg a.i./cm^2^ for surface treatments [44]. For this purpose, 1 mL of the aqueous solution that contained the appropriate volume of chlorfenapyr was sprayed on the concrete surfaces as a fine mist using an AG-4 airbrush (Mecafer S.A., Valence, France). Control dishes were sprayed with distilled water with a different AG-4 airbrush that is reserved for treatments dealing with controls. 

To obtain eggs, 100 female *T. castaneum* adults, 7 days old, were taken from the colony and placed into a 1-L glass jar filled with 500 mL flour for one day at 30 °C, 65% relative humidity and continuous darkness. Sex determination was conducted on the basis of morphological characters of adults as proposed by Halstead [48]. Twenty four hours later, adults and eggs were isolated from the flour with a No 20 (0.85 mm openings) and a No 60 (0.25 mm openings) U.S. standard testing sieves (Advantech Manufacturing Inc., New Berlin, WI, USA), respectively.

With the use of a fine brush (Cotman 111 No 000, Winsor and Newton, London, UK), 116 isolated eggs were left very carefully on 116 control dishes that contained 0.5 g of flour sprinkled over the concrete surface. Each lid had a central 1.5 cm circular opening covered with muslin gauze that allowed aeration of their internal spaces. 

A total of 160 eggs were placed very carefully on 160 treated dishes as above with a different fine brush (Cotman 111 No 000, Winsor and Newton, London, UK). Subsequently, dishes were placed in an incubator set at 32.5 °C, 65% relative humidity and continuous darkness for the entire experimental period and observed daily under an SZX9 Olympus stereomicroscope (57× total magnification) (Bacacos S.A., Athens, Greece) in order to estimate egg hatching, as well as larval development and survival. Ventilation of dishes was conducted as described above.

### 2.4. Exposure of Larvae

Eggs were obtained as described above. A total number of 160 eggs were very carefully transferred on 160 untreated dishes with a fine brush (Cotman 111 No 000, Winsor and Newton, London, UK) that contained 0.5 g of flour sprinkled over the concrete surface. Then, dishes were placed in an incubator set at 32.5 °C, 65% relative humidity and continuous darkness and inspected daily under an SZX9 Olympus stereomicroscope (57× total magnification) (Bacacos S.A., Athens, Greece) for egg hatching. Newly emerged larvae were very carefully separately transferred with a different fine brush (Cotman 111 No 000, Winsor and Newton, London, UK) to treated dishes which contained 0.5 g of flour spread as above. Dishes were inserted in an incubator set at 32.5 °C, 65% relative humidity and continuous darkness for the entire experimental period. Development and survival of larvae was estimated daily. All dishes allowed aeration as described above.

### 2.5. Exposure of Parental Adults

In this trial, 100 female adults, 7 days old, were obtained from a colony and released on treated concrete surfaces of dishes without food for three days. Then, individuals were transferred to 1-L glass jars filled with 500 mL flour at 30 °C, 65% relative humidity, and continuous darkness and left for 24 h. Next day, adults and eggs were separated with sieves as described above. A total of 164 eggs were very carefully left on 164 untreated concrete surfaces of dishes that contained 0.5 g of flour as described above. Next, dishes were placed in an incubator set at 32.5 °C, 65% relative humidity, and continuous darkness and were observed daily under an SZX9 Olympus stereomicroscope (57× total magnification) (Bacacos S.A., Athens, Greece) for egg hatching, developmental duration and survival of larvae and pupae. All dishes were aerated as above. When insects became adults, pairs were formed and kept separately in the petri dishes. Longevity of adults was recorded every 24 h. Fecundity was evaluated by calculating the number of eggs laid per female per day.

### 2.6. Statistical Analyses

Data on egg hatching, larval and pupal development, as well as adult longevity were subjected to the Shapiro–Wilk normality test, which indicated departure from a normal distribution. Therefore, data were analyzed by Kruskal–Wallis analysis of variance on ranks (Dunn’s test at *α* = 0.05). The Kaplan–Meier method [49] was used to estimate *T. castaneum* survival curves at each of the examined treatment. As the log rank test indicated that survival curves were significantly different, we used the Holm–Sidak test to determine which pairs of curves were different. All survival analyses were conducted using the SigmaPlot 14.0 [50].

The net reproductive rate $R_{0}=\sum \left(l_{x}\times m_{x}\right)$ (*l_x_* corresponds to the cohort survival to age *x* and *m_x_* the age specific fecundity), the intrinsic rate of increase (*r_m_*) $\sum (e^{r_{m}\times x}\times l_{x}\times m_{x})=1$, the finite rate of increase $\lambda =e^{r_{m}}$, the mean generation time $T=\frac{\ln R_{0}}{r_{m}}$ and the doubling time $DT=\frac{\ln2}{r_{m}}$ were calculated according to Carey [8]. Significant differences between the demographic parameters at each of the treatments were tested with the superposition of 95% confidence intervals (C.I.) (Wald test), which were obtained by bootstrapping in R [51]. In particular, for each treatment we sampled ten thousand individuals in order to derive 95% C.I.

## 3. Results

Exposure of eggs and larvae to chlorfenapyr was detrimental for *T. castaneum*, and they did not complete their development. When eggs were exposed to chlorfenapyr, 65.0% were hatched while a small percentage (14.4%) of L1 larvae developed to the L2 stage. When newly emerged larvae were exposed to chlorfenapyr, 21.9% were developed to L2 stage while only few larvae (6%) emerged to the L3 stage. The survival analysis indicated that the exposure of different developmental stages of *T. castaneum* to chlorfenapyr affected the survival probability among treatments (*x*^2^ (Log rank) = 102.922; *DF* = 3; *p* < 0.001; Figure 1 and Figure 2), as well as the mean survival time and the fecundity of progeny (when parental female adults were exposed to chlorfenapyr—see materials and methods) (Table 1). Mean survival time was significantly lower when eggs or larvae were exposed to chlorfenapyr (6.4 and 6.9 days, respectively) compared to control treatment (67.8 days) and the progeny of females which were exposed to chlorfenapyr (82.2 days). In addition, mean fecundity did not differ between control treatment (67.8 females/female) and when female progeny was exposed to chlorfenapyr (82.2 females/female).

When parental females were exposed to chlorfenapyr, the biological features of progeny did not differ significantly compared to control treatment (Table 2). Therefore, the time period for egg hatching, larval and pupal developmental periods, female and male longevities for the control treatment and the progeny of the females that were exposed to chlorfenapyr were 4.66 and 4.76 days, 25.85 and 25.71 days, 5.00 and 5.26 days, 87.33 and 104.22 days, and 76.87 and 91.87 days, respectively.

The calculated demographic parameters did not significantly differ between the control treatment and the progeny of the females which were exposed to chlorfenapyr (Table 3). The mean values of the net reproductive rate, the intrinsic rate of increase, the mean generation time and the doubling time for the control treatment and the progeny of the parental females which were exposed to chlorfenapyr were 14.3 and 9.3 females/female, 0.038 and 0.028 females/female/day, 1.039 and 1.029, 70.0 and 76.9 days, and 18.5 and 24.9 days, respectively.

## 4. Discussion

Our study provides a comprehensive description of the survival and reproductive schedules of *T. castaneum* when different developmental stages were exposed to chlorfenapyr. The knowledge of the developmental biology and the life table parameters of *T. castaneum* also facilitate the estimation of its population growth through time and therefore its potential outbreak [4,5,6,11]. All scenarios we tested are reliable since insect individuals or their eggs may be transferred during cleaning procedures [52], e.g., from treated to untreated areas and vice versa or between areas that are both treated or untreated. Our results provide evidence that the exposure of different developmental stages to chlorfenapyr is associated with differences in the performance of *T. castaneum*. Tabulating the survivorship and fecundity schedules of individuals from birth to death, we showed that *T. castaneum* did not complete development when eggs or neonate larvae were exposed to chlorfenapyr. We also found that the exposure of eggs and neonate larvae to the insecticide was harmful for the population evolution since the overall emerged population collapsed in 9 (case of exposed eggs) or 11 days (case of exposed neonate larvae). In addition, the mean survival time, as well as the survival curves of the cohort whose eggs were exposed to the insecticide, differed significantly compared to the one that only larvae were exposed. A careful inspection of the complete survival curves reveals the nature of this disparity. In particular, it is immediately apparent that the two curves are virtually identical for the initial half of their course. After five days approximately, more individuals of the larvae-exposed to chlorfenapyr cohort survived, a deduction that explains with clarity the difference in the log-rank test.

The finding that the early exposure of eggs and newly emerged larvae of *T. castaneum* to chlorfenapyr treated concrete surfaces did not allow the completion of the biological cycle is important for the evaluation of chlorfenapyr against stored-product pests, as it accounts for effects on the population level of the target insects. It should be noted that the exposure was conducted with the presence of flour as a food source. Food can partially absorb chlorfenapyr leading to the reduction of its effectiveness [53]. In contrast, when *T. castaneum* four-week larvae were exposed on surfaces partially treated with chlorfenapyr, a certain number did finally develop to adults, although several of them died after their emergence [46]. This difference could be explained by the fact that older larvae have the potential to continue their development. For example, for the same species, Sağlam et al. [54] found that old larvae (4–7 days) were more tolerant than young larvae (1–3 days) on concrete treated with chlorantraniliprole, thiamethoxam, and imidacloprid, under different combinations of biotic and abiotic conditions, since there were individuals which survived even after 14 days of exposure. A similar trend has been postulated for old vs. young larvae of other stored-product insects exposed to chlorfenapyr, i.e., *T. granarium* [42] and the yellow mealworm beetle, *Tenebrio molitor* L. (Coleoptera: Tenebrionidae) [55]. In a recent study, Thorat et al. [56] reported that the application of sublethal doses of chlorfenapyr to a mixture of whole wheat flour with 5% Brewer’s yeast allowed the development of exposed eggs of *T. castaneum* to adults and consequently the continuation of infection of flour. Our study suggests that the ovicidal effect of chlorfenapyr becomes a crucial issue for the management of *T. castaneum*. Thorat et al. [56] also reported that chlorfenapyr negatively affects the viability of eggs of *T. castaneum* depending on the applied sublethal dose. From a practical point of view, the application of chlorfenapyr on surfaces will not affect only eggs of *T. castaneum* since it is capable of killing eggs of other stored-product insects. For example, in a recent study, Boukouvala and Kavallieratos [44] found that the exposure of *T. granarium* eggs on chlorfenapyr-treated concrete resulted in a maximal 87 and 76.7% hatching with the presence of food or not, respectively. In both cases, the emerged larvae did not complete their development, correspondingly to the findings of the current study, albeit we did not examine the exposure of eggs without the presence of flour. The insertion of flour, however, extended the survival period of *T. granarium* larvae on concrete treated with chlorfenapyr given that it was observed 100% mortality eight days post-treatment with flour vs. five days without flour [44]. One other important factor that regulates the survival of the exposed insects is the implemented dose of insecticide. In an earlier study, Arthur [41], by treating concrete surfaces with chlorfenapyr solutions that corresponded to label but also to lower doses (i.e., 0.0275–0.0825 mg a.i./cm^2^), found that the absence of food per dose reduced the survival of *T. castaneum* adults on concrete surfaces. Whether *T. castaneum* larvae perform similarly when food is absent, under our experimental approach, merits further investigation.

We showed that exposure of parental adult females to chlorfenapyr did not affect the developmental biology of progeny. Thus, egg hatching, developmental duration of larvae and pupae, female and male longevity, as well as fecundity, did not differ significantly compared to the control treatment. Furthermore, the survival analysis indicated that there were no statistically significant differences between the survival curves and the mean survival times of these treatments. These facts are also depicted on the values of the estimated life table parameters. The net reproductive rate, the intrinsic and finite rates of increase, the mean generation time and the doubling time between control treatment and the progeny of the exposed parental females to chlorfenapyr, did not differ significantly. These results are biologically interpretable, since life table parameters depend on cohort survival, development and fecundity [4]. Therefore, there is no effect on the population growth of *T. castaneum* when parental females are exposed to chlorfenapyr. It should be noted that life table statistics rely on several assumptions [11,47]. For instance, the calculation of the life table parameters assumes that the studied population is closed, exhibiting constant birth and death rates and exponential population increase [11]. Although these situations are rarely met in nature, we expect that the calculated life table parameters are indicative of *T. castaneum* potential population increase.

*Tribolium castaneum* is a flyer [19,57,58] and it colonizes the available food sources by walking and flying [57,58]. It also exhibits high dispersive behavior during adult stage through flight [57,58]. This means that the presence of adults should trigger meticulous management tactics including surface treatments with chlorfenapyr in storage facilities since the offspring production will continue their normal development to the next generation even if the parental female adults come in contact with treated surfaces but move away from them afterwards, as we examined here. Apart from stored-commodities or floury surfaces, *T. castaneum* is found in cracks and crevices that contain food residues [59,60] where it may oviposit, although less likely [21]. However, insecticidal applications should be implemented in combination with thorough sanitation procedures to maximize the control efforts against *T. castaneum* [50].

## 5. Conclusions

In the light of our experiments, we expect the results of this study to have bearing on the management of *T. castaneum*, as they reveal its performance after treatments with chlorfenapyr. We showed that the exposure of different developmental stages to this insecticide is associated with the differences recorded on fitness components of *T. castaneum*. Furthermore, biological features and demographic parameters of *T. castaneum* may be incorporated into population models evaluating the population dynamics of this species, as well as mass-rearing models allowing for efficient breeding in the insectary [11,61]. Eggs and newly emerged larvae are key instars for the effective management of *T. castaneum*. Given that chlorfenapyr is currently registered for structural treatments in storage facilities, we expect that further studies will take into account our approach for the evaluation of additional registered insecticides or novel a.i. as grain protectants against *T. castaneum* and other stored-product species.

## Figures and Tables

**Figure 1 insects-11-00542-f001:**
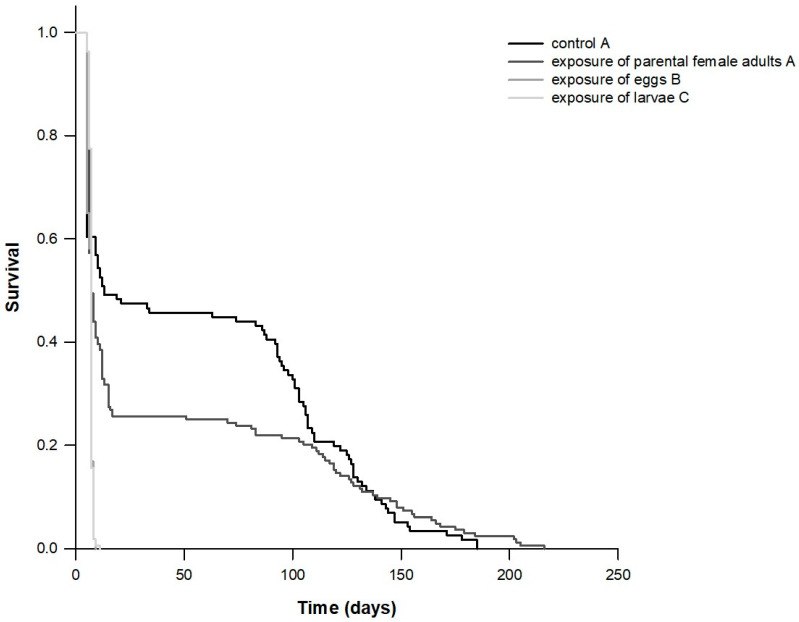
Survival curves of cohorts of *Tribolium castaneum* eggs, larvae, and progeny of parental female adults exposed to concrete treated with chlorfenapyr. The survival curves of the treatments followed by the same letter are not statistically different.

**Figure 2 insects-11-00542-f002:**
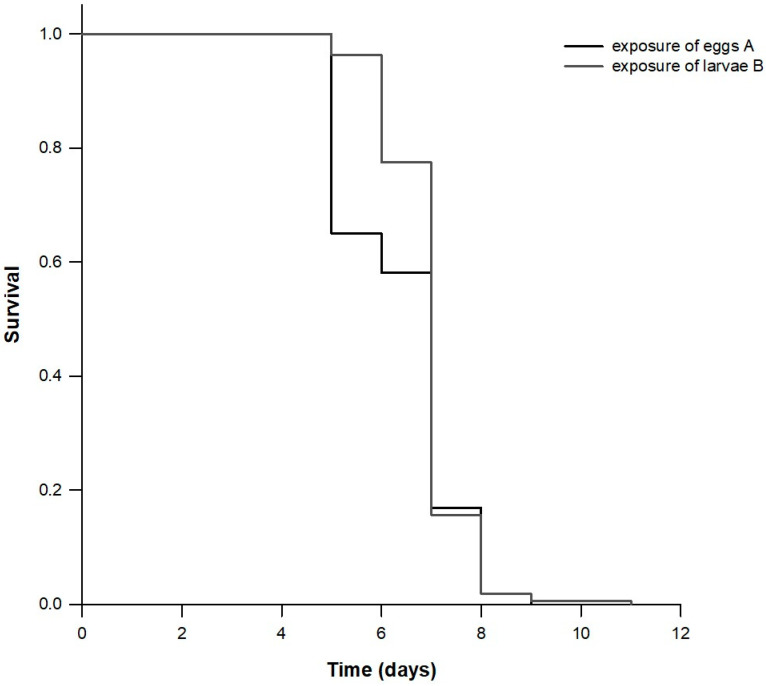
Magnification of survival curves of cohorts of *Tribolium castaneum* eggs and larvae exposed to concrete treated with chlorfenapyr. Different letters indicate that curves are statistically different.

**Table 1 insects-11-00542-t001:** Mean survival times and fecundity (95% C.I.) of *Tribolium castaneum*.

Treatment	Survival Time (Days)	95% C.I.	Fecundity (Females/Female)	95% C.I.
Control	57.8 A	47.2–68.5	67.8 A	38.2–97.4
Exposure of eggs	6.4 B	6.2–6.6	-	-
Exposure of larvae	6.9 C	6.8–7.0	-	-
Exposure of parental female adults	39.7 A	30.8–48.6	82.2 A	45.4–119.0

Means within a column followed by the same letter are not statistically different. Where dashes exist, no fecundity was observed.

**Table 2 insects-11-00542-t002:** Duration of developmental stages and adult longevity in days (mean ± SE, median) of *Tribolium castaneum* when female parental adults were exposed to chlorfenapyr.

Treatment	Egg	Larva	Pupa	Female	Male
Control	4.66 ± 0.08	25.85 ± 0.30	5.00 ± 0.07	87.33 ± 5.45	76.87 ± 4.69
	5.0 A	25.0 A	5.0 A	90.0 A	71.0 A
Exposure of parental female adults	4.76 ± 0.07	25.71 ± 0.51	5.26 ± 0.10	104.22 ± 7.68	91.88 ± 8.88
	5.0 A	25.0 A	5.0 A	98.0 A	82.0 A
*H*	1.163	0.985	3.474	522.500	1.770
*DF*	1	1	1	1	1
*p*	0.281	0.321	0.062	0.084	0.183

Medians within a column followed by the same letter are not statistically different (Kruskal–Wallis analysis of variance on ranks, Dunn’s test at *a* = 0.05).

**Table 3 insects-11-00542-t003:** Values of net reproductive rate (*R*_0_), intrinsic rate of increase (*r_m_*), finite rate of increase (*λ*), mean generation time (*T*), and doubling time (*DT*) of *Tribolium castaneum* (mean, 95% C.I.) when parental female adults were exposed to chlorfenapyr.

Treatment	Net Reproductive Rate (Females/Female) $R_{0}=\sum \left(l_{x}\times m_{x}\right)$		Intrinsic Rate of Increase (Females/Female/Day) $\sum \left(e^{r_{m}\times x}\times l_{x}\times m_{x}\right)=1$		Finite Rate of Increase $\lambda =e^{r_{m}}$		Mean Generation Time (Days) $T=\frac{\ln R_{0}}{r_{m}}$		Doubling Time (Days) $DT=\frac{\ln2}{r_{m}}$	
	Mean	95% C.I.	Mean	95% C.I.	Mean	95% C.I.	Mean	95% C.I.	Mean	95% C.I.
Control	14.3 A	9.7–18.7	0.038 A	0.032–0.043	1.039 A	1.032–1.044	70.0 A	65.7–74.3	18.5 A	16.1–22.0
Exposure of parental female adults	9.3 A	4.6–14.4	0.028 A	0.020–0.035	1.029 A	1.020–1.036	76.9 A	69.1–88.0	24.9 A	19.8–34.7

Means within a column followed by the same letter are not statistically different.

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
