# Peer review of "How Is Fitness of Tribolium castaneum (Herbst) (Coleoptera: Tenebrionidae) Affected When Different Developmental Stages Are Exposed to Chlorfenapyr?"

_insects, 2020, doi:10.3390/insects11080542_

Round 1
Reviewer 1 Report
This is a very interesting study of a widely used pro-insecticide. The authors demonstrate convincingly that adult females in the circumstances of the experiment are not affected by the treatment. Since this pro-insecticide is often applied in practice when adults are noticed in cracks etc., the findings are of wide interest.
An earlier study cited and discussed by the authors noted that chlorfenapyr was lethal to two species of Tribolium adults. That study sprayed a given dose on concrete and then held adult beetles for 7 days without food. The paper would be improved if the authors discussed in more detail two possible differences between their study and the earlier one: (1) the concentration of chlorfenapyr used (I believe they were similar but this paper states it used the recommended dosage; while the earlier paper (Arthur 2008 J Stored Products Research) provided information on the mixture used and its concentration.) and (2) the authors note that 'with food' vs 'without food' can be an important consideration in effectiveness of insecticides in general. I would like a bit more discussion of this point as it may be responsible for the different results of the two papers.
Author Response
We thank Reviewer 1 for the very positive comments about the content of our study. It is highly appreciated from our side.
We enhanced the Discussion section according the suggestions of Reviewer 1. Please see lines 255-263.
Reviewer 2 Report
The manuscript entitled: How is fitness of Tribolium castaneum (Herbst) (Coleoptera: Tenebrionidae) affected when different developmental stages are exposed to chlorfenapyr"
deals with the effect of the insecticide chlorfenapyr on the population fitness of the red flour beetle, Tribolium castaneum, generally regarded as a serious pest of a large spectrum of stored products. The MS is well written as well as the experimental design is properly planned. It is my beliefe that the obtained results can represent an interesting tool, to be considered within the IPM programs for the management of the infestations of this insect pest.
Author Response
We thank Reviewer 2 for the very positive comments about the content of our study. It is highly appreciated from our side.